# Optimising Cancer Medicine in Clinical Practices: Are Neoadjuvant and Adjuvant Immunotherapies Affordable for Cancer Patients in Low- and Middle-Income Countries?

**DOI:** 10.3390/cancers17101722

**Published:** 2025-05-21

**Authors:** Rashidul Alam Mahumud

**Affiliations:** 1NHMRC Clinical Trials Centre, Faculty of Medicine and Health, The University of Sydney, Camperdown, NSW 2006, Australia; rashed.mahumud@sydney.edu.au; Tel.: +61-452457242; 2School of Business, Faculty of Business, Education, Law and Arts, University of Southern Queensland, Toowoomba, QLD 4350, Australia; 3Centre for Health Research, University of Southern Queensland, Toowoomba, QLD 4350, Australia

**Keywords:** neoadjuvant, adjuvant immunotherapies, cancer, low- and middle-income countries

## Abstract

Although cancer immunotherapy before or after surgery has dramatically improved outcomes in high-income countries, its high cost and the need for specialised facilities often put these treatments out of reach for patients in low- and middle-income countries. This perspective examines how giving immunotherapy around the time of surgery can help shrink tumours and prevent relapses and explores the financial, logistical, and workforce challenges faced in resource-limited settings. By proposing policy changes, local manufacturing partnerships, and targeted funding strategies, this work aims to guide healthcare leaders and global stakeholders in making these life-saving therapies affordable and accessible to all cancer patients worldwide.

## 1. Introduction

Over recent decades, cancer has become a leading cause of mortality worldwide, with low- and middle-income countries (LMICs) disproportionately affected [1]. Technological advancements in genomics, immunology, and molecular biology have led to the development of novel therapeutic strategies that have significantly altered the landscape of cancer care. Among these innovations, immunotherapy has demonstrated remarkable clinical benefits and survival outcomes in cancers such as melanoma, non-small cell lung cancer (NSCLC), renal cell carcinoma, and other malignancies [2,3,4]. Specifically, neoadjuvant (pre-operative) and adjuvant (post-operative) immunotherapies have garnered significant attention due to their ability to downstage tumours prior to surgical intervention and eliminate residual disease following curative-intent treatments, thereby improving clinical outcomes [5].

However, despite these promising advances, a critical concern persists regarding whether or not such innovative treatments are practically accessible and affordable for cancer patients in LMICs. Moreover, accessibility challenges are not confined to LMICs; even in many high-income countries, escalating drug prices, insurance coverage gaps, and growing co-payment burdens can limit patient access to life-saving immunotherapies. This concern is expanded by the already existing healthcare inequities characterised by limited resources, financial constraints, and infrastructural inadequacies commonly observed in these settings. Although considerable research has documented the effectiveness of immunotherapy, limited attention has been paid to strategies aimed at bridging the affordability gap in LMICs. Global cancer statistics highlight stark disparities in health equity, where patients in resource-constrained settings are often unable to benefit from high-cost, advanced treatments due to systemic financial and logistic barriers [1,6,7].

Moreover, recent projections suggest that the cancer burden in LMICs will increase substantially over the next two decades, further compounding the urgency of making effective treatments financially viable in these contexts. The lack of integration of pharmacoeconomic evidence into health policy frameworks and reimbursement systems in LMICs further limits the uptake of immunotherapies. Without clear cost-effectiveness thresholds, decision makers struggle to justify investment in high-cost interventions, even when clinical efficacy is well established. Therefore, addressing affordability requires a dual approach—strengthening the economic evaluation capacity and ensuring global pricing reforms tailored to LMIC contexts.

One of the primary challenges facing the integration of immunotherapies into clinical practices in LMICs is their exceedingly high cost. Immunotherapeutic agents, particularly immune checkpoint inhibitors such as pembrolizumab, nivolumab, and atezolizumab, are among the most expensive anticancer medications available today [8]. The prohibitive cost of these therapies frequently surpasses the per capita healthcare budgets of many LMICs, thereby severely restricting their use and availability in these countries. Furthermore, many patients in LMICs face out-of-pocket expenses due to insufficient health insurance coverage and inadequate social protection mechanisms. Consequently, the majority of cancer patients are either unable to initiate or sustain immunotherapy treatments, leading to compromised clinical outcomes and poorer survival rates compared to patients in high-income countries (HICs) [9].

Additionally, infrastructural limitations in LMIC healthcare systems exacerbate the challenges of implementing immunotherapy protocols. Immunotherapy requires specific resources such as specialised diagnostic equipment for biomarker testing, adequately trained healthcare professionals, robust patient monitoring, and efficient adverse event management systems. Unfortunately, many LMICs are characterised by significant shortages in the healthcare workforce and inadequacies in laboratory and the necessary diagnostic facilities to identify suitable patient populations and to effectively manage treatment-related toxicities. Moreover, logistic barriers, including unreliable supply chains, inconsistent drug availability, and inadequate facilities for drug storage and administration, further hinder the integration of these innovative therapies into routine clinical care.

Addressing these challenges necessitates the development of targeted policy strategies, collaborations, and global health partnerships. Potential approaches include advocating for differential pricing and negotiated access agreements with pharmaceutical companies to lower the financial burden on healthcare systems in LMICs [6,10]. Additionally, the establishment of public–private partnerships could facilitate technology transfer, knowledge exchange, capacity building, and infrastructure improvement, thereby enhancing diagnostic and therapeutic capabilities in resource-limited settings. Efforts to integrate immunotherapy into universal health coverage schemes and strengthen national cancer control programs can also improve equitable patient access and affordability. Further, international donor agencies and global health stakeholders must prioritise funding allocation for cancer care, with a specific focus on expanding immunotherapy access and affordability in LMICs.

Hence, this perspective aims to systematically explore the clinical significance of neoadjuvant and adjuvant immunotherapies, comprehensively examine the cost-related challenges in LMICs, and propose pragmatic policy interventions to enhance their feasibility and affordability in these regions. By highlighting and addressing these critical economic and infrastructural barriers, the global oncology community can support the equitable dissemination and adoption of life-saving cancer therapies, ensuring that cancer patients in LMICs are no longer excluded from benefiting from advances in immunotherapy.

## 2. Current Landscape of Licensed Anticancer Drugs

The data are presented as both absolute numbers and percentages relative to the total number of licensed anticancer drugs (Table 1). Out of 321 licensed anticancer drugs, 290 (90.34%) have received approval from the U.S. Food and Drug Administration (FDA), indicating that the vast majority of these drugs meet the rigorous safety and efficacy standards required by the FDA. In addition, 215 drugs (66.98%) have obtained approval from the European Medicines Agency (EMA), reflecting partial regulatory alignment but also a notable 23.36 percentage-point gap compared to the FDA. Furthermore, approvals by the EMA often lag behind FDA decisions by one to three years on average, delaying European patient access to new therapies. Moreover, 50 drugs (15.58%) are approved at a national level within Europe—reflecting approvals by individual European countries’ regulatory bodies—while a small subset of 11 drugs (3.43%) has been approved by agencies outside of the FDA, EMA, or European national authorities, suggesting alternative regulatory pathways in other regions.

This global diversity in regulatory approval reflects both the opportunities and challenges in ensuring equitable access to anticancer drugs across jurisdictions. In LMICs, regulatory harmonisation and timely approval often lag due to limited infrastructure and resource constraints, thereby delaying access to potentially life-saving treatments. Additionally, the lack of local manufacturing and dependency on imported pharmaceuticals can exacerbate these delays, especially for newly developed or patented agents. While the high FDA and EMA approval rates indicate robust evidence-based evaluation, such approvals do not automatically translate into affordability or inclusion in national formularies in LMICs. Therefore, regulatory approval status, while necessary, is not sufficient for widespread access in global oncology.

Notably, 54 of the licensed anticancer drugs (16.82%) are included in the World Health Organization’s Essential Medicines List (WHO EML). The inclusion in the WHO EML is particularly significant because it identifies drugs that are deemed critical for addressing key health needs, especially in resource-constrained settings. Furthermore, the table categorises the drugs based on their formulation. An overwhelming majority of 317 drugs (98.75%) are formulated as single active pharmaceutical ingredient (API) therapies, while only 4 drugs (1.25%) are combination therapies containing more than one API. This distribution indicates that the vast majority of anticancer drugs on the market are designed as single-agent therapies rather than multi-component combinations. These data illustrate the extensive regulatory coverage of anticancer drugs across major agencies, underscore the predominance of single-agent formulations, and highlight the importance of a select group of these drugs as essential medicines according to WHO criteria.

## 3. Clinical Significance of Neoadjuvant and Adjuvant Immunotherapies

Immunotherapies target immune checkpoints such as programmed cell death receptor-1 (PD-1) and its ligand (PD-L1), as well as cytotoxic T-lymphocyte-associated antigen-4 (CTLA-4) [8]. By inhibiting these immune checkpoints, immune cells regain their capacity to identify and eliminate tumour cells. In a neoadjuvant setting, immunotherapy effectively induces tumour regression before surgical resection, potentially resulting in improved surgical outcomes and enhanced pathological response rates [9,11]. Moreover, this approach can reduce tumour burden significantly, allowing for less extensive surgery, fewer complications, and improved post-operative recovery (Figure 1).

Neoadjuvant immunotherapy offers several notable clinical advantages. Firstly, it enables real-time assessment of treatment response through a pathological analysis of surgical specimens, facilitating early and informed decisions regarding subsequent therapeutic strategies. Patients who achieve a pathological complete response (pCR)—characterised by the absence of viable tumour cells upon histological examination—generally experience significantly better long-term survival outcomes compared to those who do not achieve pCR. For instance, recent clinical trials have demonstrated notable pCR rates ranging from 30% to 60% among patients with NSCLC receiving neoadjuvant immune checkpoint inhibitors, leading to improved disease-free survival and reduced rates of tumour recurrence [12]. Furthermore, neoadjuvant immunotherapy can stimulate systemic immune responses, providing potential immunological memory against tumour antigens and thereby reducing the likelihood of metastatic relapse after surgical removal of the primary tumour.

Adjuvant immunotherapy, administered post-surgery or radiotherapy, aims to eliminate residual tumour cells and micro-metastases, thus substantially decreasing recurrence rates and extending disease-free survival [13,14]. This therapeutic strategy is particularly beneficial in high-risk patients who may harbour microscopic residual disease undetectable through conventional imaging modalities or standard histopathological techniques. Adjuvant immunotherapy thus offers a critical opportunity to eradicate subclinical tumour deposits that would otherwise progress to metastatic disease. Notably, several landmark trials have shown remarkable outcomes with adjuvant immunotherapy in various cancers. For example, in advanced melanoma, the use of adjuvant PD-1 inhibitors such as nivolumab and pembrolizumab has substantially lowered recurrence risk, extending both disease-free survival and overall survival compared to traditional chemotherapy or observation alone [15]. Similarly, the KEYNOTE-045 trial demonstrated that pembrolizumab improved overall survival versus chemotherapy in advanced urothelial carcinoma [16], and the CheckMate 025 trial established superior survival with nivolumab over everolimus in metastatic renal cell carcinoma [17]. These findings support the widespread adoption of adjuvant immunotherapy in clinical practice guidelines for high-income settings.

The clinical benefits associated with neoadjuvant and adjuvant immunotherapies have rapidly translated into regulatory approvals and routine clinical adoption across numerous high-income countries, reshaping cancer care paradigms significantly. However, it is important to recognise the disparity that exists between high-income countries (HICs) and LMICs in terms of immunotherapy adoption. The current evidence supporting immunotherapy predominantly originates from clinical trials conducted in HICs, often involving cancers such as melanoma, NSCLC, renal cell carcinoma, and urothelial carcinoma, which have lower prevalence rates in LMIC settings.

In contrast, evidence for immunotherapy’s effectiveness in malignancies prevalent in LMICs, including cervical, liver, gastric, and head and neck cancers, remains limited. Many cancers frequently observed in LMIC populations are associated with infectious agents or unique epidemiological risk factors and thus may exhibit distinct tumour biology, immune microenvironments, and treatment responses compared to cancers prevalent in HICs. Emerging studies suggest that immunotherapy may indeed have significant potential efficacy in these LMIC-specific malignancies. For instance, preliminary findings in hepatocellular carcinoma, a leading cause of cancer mortality in many LMICs, indicate promising responses to PD-1 inhibitors in advanced stages [18,19,20,21,22]. Similarly, early-phase trials have suggested that cervical cancer, a common malignancy in resource-limited settings, may benefit from combination immunotherapies, especially when integrated with standard chemotherapy or radiotherapy [21,22].

Despite these promising findings, extensive clinical studies specifically conducted in LMIC populations remain sparse, hindering the development of tailored, region-specific therapeutic guidelines. Conducting robust clinical trials within LMIC contexts is essential to adequately evaluate immunotherapy’s effectiveness and safety profile, taking into consideration genetic, environmental, and socioeconomic factors unique to these populations. Therefore, generating rigorous, contextually relevant evidence is critical to expanding immunotherapy access and optimising cancer care outcomes in LMICs.

## 4. Cost and Affordability Challenges

Immune checkpoint inhibitors rank among the most costly cancer treatments globally, typically costing tens of thousands of dollars annually per patient [10,23,24]. While these high prices partially reflect extensive research and development, complex manufacturing processes, and intellectual property considerations, they are also driven by value-based pricing strategies that estimate the maximum price patients or healthcare systems are willing to pay for demonstrated survival gains and quality-of-life improvements. In LMICs, where healthcare financing heavily depends on out-of-pocket expenditures and public healthcare systems are chronically underfunded, these costs present an insurmountable barrier to treatment accessibility [10]. Importantly, similar affordability issues arise in many HICs, where substantial co-payments, insurance coverage gaps, and restrictive formularies prevent some patients from accessing these therapies despite high national incomes.

Beyond drug costs, systemic challenges such as inadequate diagnostic imaging, shortages of trained oncology specialists, and limited laboratory infrastructure further compound affordability and accessibility issues. Biomarker assessments (such as PD-L1 expression and tumour mutational burden) are essential for patient selection and optimising treatment efficacy, yet remain inaccessible due to infrastructural deficiencies in many LMIC settings [25,26]. Consequently, scarce resources risk being misallocated, depriving eligible patients of potentially life-saving therapies, while inadvertently promoting inequities in cancer care delivery.

Furthermore, many LMICs lack robust health technology assessment (HTA) frameworks to guide reimbursement and policy decisions, contributing to inconsistent pricing and fragmented procurement processes. Even in HICs with established HTA bodies, differing WTP thresholds and budget constraints can delay or limit reimbursement approvals for high-cost immunotherapies. Without transparent pricing models or value-based assessment systems, cancer medicines remain unaffordable and are often excluded from national essential medicines lists. Even when treatments are subsidised, logistical barriers—such as inadequate cold chain storage, irregular drug supply, and transportation difficulties—further restrict access to timely immunotherapy. These layered challenges make cost containment and policy innovation vital to closing the affordability gap.

The affordability gap is exacerbated by socioeconomic determinants of health, geographical isolation, and uneven distribution of specialised oncology centres within LMICs. Patients from marginalised backgrounds and remote areas frequently encounter severe barriers to treatment access, perpetuating disparities and undermining health equity goals [25,26].

## 5. Strategies to Improve Accessibility and Affordability

Effective price negotiation strategies have demonstrated success in various contexts but require supportive regulatory environments to ensure equitable drug distribution [27]. Regional or global pooled procurement mechanisms represent a promising strategy to lower drug costs significantly by leveraging collective bargaining power [28]. Additionally, national healthcare systems could implement innovative financial solutions, including cancer-specific insurance schemes, targeted subsidies, or public–private partnerships that enhance both treatment affordability and infrastructure capacity [29].

Local production of immunotherapeutic agents through technology transfer agreements with pharmaceutical companies could substantially lower costs, mitigate international supply chain disruptions, and foster economic self-sufficiency within LMICs [30]. Although establishing manufacturing facilities demands initial investment, the long-term economic and healthcare benefits significantly outweigh the short-term costs. Global collaborative research initiatives can address critical evidence gaps by generating data relevant to LMIC populations. While participation in clinical trials can offer enrolled patients temporary access to immunotherapies, such mechanisms are time-limited, restricted to selected sites, and do not guarantee affordability or availability for the broader patient population. Accordingly, clinical trials should complement—but not replace—sustainable financing and procurement strategies. Moreover, these trials enhance the scientific community’s understanding of immunotherapy efficacy in diverse patient cohorts, shaping context-specific clinical guidelines. In high-income countries like Australia, the integration of precision medicine technologies—including comprehensive genomic profiling, three-dimensional imaging, and AI-enabled diagnostics—has improved early detection and personalised treatment planning, particularly for melanoma, demonstrating the potential scalability of such innovations [31]. Incorporating these approaches into national cancer control plans could improve outcomes and resource efficiency.

Importantly, providing early-stage patients with neoadjuvant or adjuvant immunotherapy is demonstrably more cost-effective than deploying the same agents in metastatic disease: for example, adjuvant pembrolizumab in resected stages IIB/IIC melanoma has an Incremental cost-effectiveness ratio (ICER) of CHF 27,424 (EUR 27,342) per quality-adjusted life year (QALY) versus observation [32], whereas treatment of BRAF-mutant metastatic melanoma with nivolumab or pembrolizumab yields ICERs of approximately EUR 31,137 and EUR 20,161 per QALY, respectively [33]. These differences reflect greater life years gained and shorter treatment durations in the adjuvant/neoadjuvant settings, resulting in lower overall drug and care costs. Alongside therapeutic strategies, primary prevention remains the most cost-effective long-term approach to reducing NSCLC incidence and mortality. Comprehensive smoking cessation programs, strong tobacco control policies, and air pollution reduction measures—though they yield benefits over decades—are essential to curtail the future burden of lung cancer in all income settings. Cost-effective biomarker testing strategies, such as simplified assays for PD-L1 and blood-based diagnostics, represent promising avenues to enhance patient selection accuracy while controlling costs [25,26]. Identification of predictive biomarkers for immunotherapy response and resistance can further streamline resource allocation by guiding treatment decisions towards patients most likely to benefit. Comprehensive health system strengthening, including specialised oncology training, improved diagnostic capacities, and patient monitoring infrastructure, is paramount. Sustainable integration of immunotherapy into LMIC health systems necessitates substantial investments in long-term capacity-building initiatives supported by both local governments and international donors [34].

The rapid adoption of immunotherapy presents ethical challenges, primarily the necessity to balance innovative treatment adoption with equity and financial protection for patients. Policymakers must reconcile the imperative for therapeutic innovation with ethical responsibilities to prevent exacerbating disparities. Enhanced transparency in pricing, cost-effectiveness, and clinical outcomes can facilitate informed policy decisions [35]. International organisations such as the WHO can provide normative guidance and foster collaborative frameworks for technology transfer, local manufacturing, and equitable policy reforms [36].

## 6. Recommendations

### 6.1. Implement Targeted Policy Interventions

Advocate for negotiations and global or regional pooled procurement mechanisms to lower the cost of high-priced immunotherapeutic agents.Expand cancer-specific insurance schemes and introduce targeted subsidies to reduce out-of-pocket expenses, ensuring that patients in LMICs can access these therapies without financial hardship.Develop transparent pricing and reimbursement policies that balance innovation with equitable access, guided by international organisations (e.g., the WHO).

### 6.2. Strengthen Health Infrastructure and Capacity

Invest in developing robust diagnostic facilities, including cost-effective biomarker testing (e.g., simplified PD-L1 assays or blood-based diagnostics) to accurately identify eligible patients.Increase training for oncologists, laboratory technicians, and support staff to ensure proper patient selection, monitoring, and management of immunotherapy-related adverse events.Improve logistics, including reliable drug supply chains and storage facilities, to support consistent availability of immunotherapeutic agents.

### 6.3. Promote Local Production and Technology Transfer

Adopt partnerships among governments, local industries, and international pharmaceutical companies to transfer technology and set up local production facilities, potentially reducing costs and dependency on imports.

### 6.4. Expand Collaborative Research and Clinical Trials

Initiate and support clinical trials within LMIC populations to generate context-specific evidence on immunotherapy efficacy, safety, and cost-effectiveness.Develop international collaborations that bring together researchers from HICs and LMICs, ensuring that clinical guidelines are tailored to regional epidemiological and socioeconomic conditions.

### 6.5. Incorporate Policy Guidance on Resource-Appropriate Care

Urge policymakers and guideline developers in LMICs to integrate economic criteria into national cancer control programs, prioritising interventions demonstrated to be both effective and cost-saving.Drawing on the EAGLE-FM example [37], recommend that regional health authorities consider formal health technology assessments before adopting high-cost immunotherapies or extensive surgeries.

## 7. Future Directions

To facilitate equitable access to neoadjuvant and adjuvant immunotherapies in low- and middle-income countries (LMICs), future work must pursue innovative financing and procurement models that go beyond conventional differential pricing. Transparent multi-stakeholder negotiations—bringing together governments, international organisations, and industry—should test tiered-pricing agreements and pooled regional procurement mechanisms. Parallel efforts to transfer manufacturing technology locally can build domestic capacity, reduce import dependency, and buffer against global supply disruptions.

Equally critical is investment in health system capacity. Targeted funding should expand affordable biomarker diagnostics—such as simplified PD-L1 and blood-based assays—and train multidisciplinary teams in patient selection, toxicity management, and pharmacovigilance. Strengthening supply chain logistics and establishing sustainable cold chain storage will ensure consistent drug availability outside major urban centres.

Finally, generating high-quality, region-specific evidence must be a priority. Collaborative clinical trials conducted in LMIC settings should evaluate immunotherapy regimens in cancers of the greatest local burden (e.g., cervical, hepatocellular, and gastric). Such studies will inform context-tailored guidelines and cost-effectiveness thresholds. Embedding these trials within cancer control programmes will help rapidly integrate findings into policy. By aligning innovative financing, capacity building, and rigorous local research, global stakeholders can bridge current disparities and sustainably scale life-saving immunotherapies across diverse healthcare contexts.

## 8. Conclusions

Neoadjuvant and adjuvant immunotherapies represent transformative advances in cancer medicine with the potential to significantly improve survival and reduce recurrence across diverse malignancies. However, in LMICs, the substantial financial and systemic barriers severely limit patient access. Addressing these barriers requires coordinated action by policymakers, healthcare providers, international organisations, and pharmaceutical stakeholders to achieve equitable integration of immunotherapy into global cancer care. Through concerted efforts in strategic policy making, capacity building, price negotiations, and international collaboration, LMICs can overcome existing disparities, ultimately advancing global health equity and reducing the worldwide cancer burden.

## Figures and Tables

**Figure 1 cancers-17-01722-f001:**
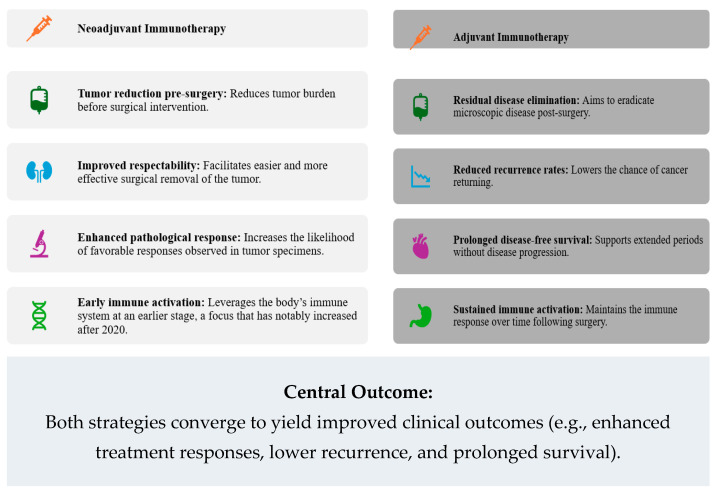
Comparative clinical benefits of neoadjuvant and adjuvant immunotherapies.

**Table 1 cancers-17-01722-t001:** This table provides a comprehensive overview of the licensing and regulatory statuses of anticancer drugs based on the latest build (as of 28 November 2024).

Summary Results as of Last Build on 28 November 2024	Number	Percentage
Number of licensed anticancer drugs	321	
FDA-approved anticancer drugs	290	90.34%
EMA-approved anticancer drugs	215	66.98%
Euro nationally approved anticancer drugs	50	15.58%
Non-FDA/EMA/European	11	3.43%
Drugs included in WHO EML	54	16.82%
Single API drugs	317	98.75%
Combination (>1 API) drugs	4	1.25%

Source: Database cancer drugs (https://www.anticancerfund.org/en/database-cancer-drugs; Accessed on 19 May 2025).

## Data Availability

Data are contained within the article. All data supporting the findings of this perspective manuscript are provided in the main text with appropriate source citations. Additional information and any further data are available from the corresponding author upon reasonable request.

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
