# Peer review of "Optimising Cancer Medicine in Clinical Practices: Are Neoadjuvant and Adjuvant Immunotherapies Affordable for Cancer Patients in Low- and Middle-Income Countries?"

_cancers, 2025, doi:10.3390/cancers17101722_

Round 1
Reviewer 1 Report
Comments and Suggestions for Authors
I have not specific criticisms or concerns about this interesting and fascinating review. I believe that is suitable for publication and of wide interest for the journal readers. The only point to be reconsidered is the length of Section 7 that suffers from redundancy due to the fact that many concepts and statements are clearly reasoned in previous sections.
Author Response
Response to the reviewer-1’s comments:
Comment 1:
I have not specific criticisms or concerns about this interesting and fascinating review. I believe that is suitable for publication and of wide interest for the journal readers. The only point to be reconsidered is the length of Section 7 that suffers from redundancy due to the fact that many concepts and statements are clearly reasoned in previous sections.
Author’s Response 1:
We thank the reviewer for their positive assessment and for highlighting that Section 7 contains content already addressed elsewhere. In response, we have substantially shortened and refocused Section 7 to eliminate redundancy and to emphasize truly novel future directions. The revised section now succinctly outlines three core areas—innovative access models, health‐system strengthening, and context-specific research—without repeating material covered in earlier sections.
The revised section reads now as follows:
“7. Future Directions
To facilitate equitable access to neoadjuvant and adjuvant immunotherapies in low- and middle-income countries (LMICs), future work must pursue innovative financing and procurement models that go beyond conventional differential pricing. Transparent multi-stakeholder negotiations—bringing together governments, international organisations, and industry—should test tiered‐pricing agreements and pooled regional procurement mechanisms. Parallel efforts to transfer manufacturing technology locally can build domestic capacity, reduce import dependency, and buffer against global supply disruptions.
Equally critical is investment in health-system capacity. Targeted funding should expand affordable biomarker diagnostics—such as simplified PD-L1 and blood-based assays—and train multidisciplinary teams in patient selection, toxicity management, and pharmacovigilance. Strengthening supply-chain logistics and establishing sustainable cold-chain storage will ensure consistent drug availability outside major urban centres.
Finally, generating high-quality, region-specific evidence must be a priority. Collaborative clinical trials conducted in LMIC settings should evaluate immunotherapy regimens in cancers of greatest local burden (e.g., cervical, hepatocellular, gastric). Such studies will inform context-tailored guidelines and cost-effectiveness thresholds. Embedding these trials within cancer control programmes will help integrate findings rapidly into policy. By aligning innovative financing, capacity-building, and rigorous local research, global stakeholders can bridge current disparities and sustainably scale life-saving immunotherapies across diverse health-care contexts.”
Please see the revised manuscript (lines 324-346).

Reviewer 2 Report
Comments and Suggestions for Authors
I think the accessibility to cancer immunotherapy in LMCI countries, and sometime even in HICs is an important topic, not enough considered and debated.
Lines 90 and 93: in my opinion, the number of Ema approved drugs, 215 vs 290 approved by FDA is not indicative of substantial regulatory concordance between US and Europe (26% lower), another point not addressed is the delay of time (sometime years) between FDA and Euro nationally approval.
Line 150: reference 16 is related to adjuvant durvalumab in NSCLC, not to urothelial carcinoma and renal cell carcinoma.
Lines180-18. I partially disagree with these statements, since the price of innovative drugs is mainly related to the estimation of the higher price that ill people are willing to pay for them, and even in many HICs there people unable to afford the treatment costs.
Lines 212: Participation in international or locally initiated clinical trials provides patients with direct access to immunotherapies otherwise financially inaccessible, thus improving both clinical outcomes and affordability. I think as the authors say above that “Conducting robust clinical trials within LMIC contexts is essential to adequately evaluate immunotherapy's effectiveness and safety profile, taking into consideration genetic, environmental, and socio-economic factors unique to these populations” however totally inadequate to solve the problem of affordability.
In my opinion, providing access to adjuvant and moreover to neoadjuvant treatments is more cost effective than providing access to immunotherapy for metastatic disease, the life-years gained are higher, and the treatment shorter. I would like a comment on this point.
A last consideration, the NSCLC incidence and mortality could be effectively reduced through educational efforts aimed at reducing smoking and pollution, and this measure, even if it takes time, is the most cost-effective nowadays.
Author Response
Response to the reviewer-2’s comments:
Comment-1:
I think the accessibility to cancer immunotherapy in LMCI countries, and sometime even in HICs is an important topic, not enough considered and debated.
Author’s response 1:
We thank the reviewer for emphasizing the global importance of immunotherapy accessibility, noting that barriers persist not only in LMICs but also in many high-income countries (HICs). The manuscript has been revised to expand the Introduction to acknowledge and briefly discuss access challenges in HICs—such as insurance coverage gaps and high out-of-pocket costs—and to frame immunotherapy affordability as a truly global concern.
These changes appear in the second paragraph of the revised Introduction, with added text highlighted for clarity. The revised text reads as follows:
“Moreover, accessibility challenges are not confined to LMICs; even in many high-income countries, escalating drug prices, insurance coverage gaps, and growing co-payment burdens can limit patient access to life-saving immunotherapies.”. Please see lines 59 to 61.
Comment-2:
Lines 90 and 93: in my opinion, the number of Ema approved drugs, 215 vs 290 approved by FDA is not indicative of substantial regulatory concordance between US and Europe (26% lower), another point not addressed is the delay of time (sometime years) between FDA and Euro nationally approval.
Author’s response 2:
Thanks. The reviewer pointed out that a 26% difference between FDA and EMA approvals does not necessarily indicate substantial concordance, and highlighted the importance of approval time lags. We have revised this section as:
We have modified the description of FDA vs. EMA approval rates to acknowledge the divergence in numbers.
The revised texts read as: “In addition, 215 drugs (66.98%) have obtained approval from the European Medicines Agency (EMA), reflecting partial regulatory alignment but also a notable 23.36 percentage‐point gap compared to the FDA. Furthermore, approvals by the EMA often lag behind FDA decisions by one to three years on average, delaying European patient access to new therapies.” Please see line 127 to 131.
We have added sentences to discuss the typical time delay between U.S. and European approvals and its implications for access.
The revised texts read as: “This global diversity in regulatory approval reflects both the opportunities and challenges in ensuring equitable access to anticancer drugs across jurisdictions. In LMICs, regulatory harmonisation and timely approval often lag due to limited infrastructure and resource constraints, thereby delaying access to potentially life-saving treatments. Additionally, the lack of local manufacturing and dependency on imported pharmaceuticals can exacerbate these delays, especially for newly developed or patented agents. While the high FDA and EMA approval rates indicate robust evidence-based evaluation, such approvals do not automatically translate to affordability or inclusion in national formularies in LMICs. Therefore, regulatory approval status, while necessary, is not sufficient for widespread access in global oncology.”. Please see lines 136 to 144.
Comment-3:
Line 150: reference 16 is related to adjuvant durvalumab in NSCLC, not to urothelial carcinoma and renal cell carcinoma.
Author’s response 3:
We sincerely apologise for the miscitation in our discussion of adjuvant immunotherapy. Reference 16 indeed pertains to durvalumab in NSCLC rather than urothelial carcinoma or renal cell carcinoma. We have corrected this by citing the appropriate pivotal trials for those indications.
The revised texts read as follows: “Similarly, the KEYNOTE-045 trial demonstrated that pembrolizumab improved overall survival versus chemotherapy in advanced urothelial carcinoma (16), and the CheckMate 025 trial established superior survival with nivolumab over everolimus in metastatic renal cell carcinoma(17). These findings support the widespread adoption of adjuvant immunotherapy in clinical practice guidelines for high-income settings.”. Please see lines 194-199.
References:
- Bellmunt J, de Wit R, Vaughn DJ, Fradet Y, Lee JL, Fong L, et al. Pembrolizumab as Second-Line Therapy for Advanced Urothelial Carcinoma. N Engl J Med. 2017;376(11):1015–26.
- Motzer RJ, Escudier B, McDermott DF, George S, Hammers HJ, Srinivas S, et al. Nivolumab versus Everolimus in Advanced Renal-Cell Carcinoma. N Engl J Med. 2015;373(19):1803–13.
Comment-4:
Lines180-18. I partially disagree with these statements, since the price of innovative drugs is mainly related to the estimation of the higher price that ill people are willing to pay for them, and even in many HICs there people unable to afford the treatment costs.
Author’s response 4:
We thank the reviewer for noting that the high prices of innovative cancer drugs also reflect value-based pricing—i.e. estimations of what patients or society are willing to pay for incremental clinical benefits—and that affordability challenges persist even in high-income countries (HICs). In response, we have expanded Section 4 to:
- Acknowledge the role of value-based pricing and willingness-to-pay (WTP) in setting drug prices.
The revised texts read as “While these high prices partially reflect extensive research and development, complex manufacturing processes, and intellectual property considerations, they are also driven by value-based pricing strategies that estimate the maximum price patients or healthcare systems are willing to pay for demonstrated survival gains and quality-of-life improvements.” Please see line 231-235.
- Highlight that patients in many HICs still face substantial co-payments, insurance coverage gaps, and other financial barriers.
The revised texts read as “Importantly, similar affordability issues arise in many HICs, where substantial co-payments, insurance coverage gaps, and restrictive formularies prevent some patients from accessing these therapies despite high national incomes.”. Please see lines 240-243.
………. “Furthermore, many LMICs lack robust health technology assessment (HTA) frameworks to guide reimbursement and policy decisions, contributing to inconsistent pricing and fragmented procurement processes. Even in HICs with established HTA bodies, differing WTP thresholds and budget constraints can delay or limit reimbursement approvals for high-cost immunotherapies. Without transparent pricing models or value-based assessment systems, cancer medicines remain unaffordable and are often excluded from national essential medicines lists. Even when treatments are subsidized, logistical barriers—such as inadequate cold-chain storage, irregular drug supply, and transportation difficulties—further restrict access to timely immunotherapy. These layered challenges make cost-containment and policy innovation vital to closing the affordability gap.” Please see lines 252-261.
Comment-5:
Lines 212: Participation in international or locally initiated clinical trials provides patients with direct access to immunotherapies otherwise financially inaccessible, thus improving both clinical outcomes and affordability. I think as the authors say above that “Conducting robust clinical trials within LMIC contexts is essential to adequately evaluate immunotherapy's effectiveness and safety profile, taking into consideration genetic, environmental, and socio-economic factors unique to these populations” however totally inadequate to solve the problem of affordability.
Author’s response 5:
We have revised Section 5 to clarify that, although trials can provide temporary access for participants, they do not deliver sustainable, population-wide affordability. We have de-emphasized trials as a primary affordability strategy and expanded the discussion of comprehensive financing, procurement, and policy mechanisms needed to ensure long-term access for all patients. Please see lines 268-308.
Comment-6:
In my opinion, providing access to adjuvant and moreover to neoadjuvant treatments is more cost effective than providing access to immunotherapy for metastatic disease, the life-years gained are higher, and the treatment shorter. I would like a comment on this point.
A last consideration, the NSCLC incidence and mortality could be effectively reduced through educational efforts aimed at reducing smoking and pollution, and this measure, even if it takes time, is the most cost-effective nowadays
Author’s response 6:
We have revised the Section 5 to acknowledge these points and incorporated a brief discussion of smoking‐cessation and pollution‐reduction strategies as the most cost‐effective long‐term measures for reducing NSCLC burden.
The relevant revised texts read as “Alongside therapeutic strategies, primary prevention remains the most cost-effective long-term approach to reducing NSCLC incidence and mortality. Comprehensive smoking-cessation programs, strong tobacco control policies, and air pollution reduction measures—though they yield benefits over decades—are essential to curtail the future burden of lung cancer in all income settings.” Please see lines 303-307.

Reviewer 3 Report
Comments and Suggestions for Authors
I think the topic is very important and innovative. I would like to support this topic under the quality of life cancer health disparities
However, methodologically authors need to work on more
1) methods: no mention about the method. this is narrative review. so write the section of the methods applicable as narrative review
2) main body:
Cost and Affordability Challenges : this is very brief. Please add more details including synthesize of tables, what are the existing articles regarding this topic, what are the main themes extracted by the relevant articles.
Strategies: no references at all to improve this issues. same thing, select relevant articles and create the tables, and make the theme to show the strong evidence.
Comments on the Quality of English Languagenone
Author Response
Response to Reviewer-3’s comments
Comment-1:
I think the topic is very important and innovative. I would like to support this topic under the quality of life cancer health disparities. However, methodologically authors need to work on more:
Author’s response-1:
We are grateful for the positive assessment of the topic’s importance. As a Perspective article, the manuscript’s objective is to provide an expert, forward-looking viewpoint rather than a systematic or narrative evidence synthesis; consequently, the scope and structure differ from those expected of traditional review articles.
Comment-2:
methods: no mention about the method. this is narrative review. so write the section of the methods applicable as narrative review.
Author’s response 2:
According to the Cancers Author Instructions, Perspective pieces “present a personal viewpoint on a field of research… and do not require a dedicated Methods section.”
Our manuscript therefore dispenses with formal review methods and instead draws on representative, high-impact studies, authoritative databases (e.g., WHO EML, FDA, EMA), and our own health-economics expertise to frame key affordability issues and policy options.
Comment-3:
2) main body:
Cost and Affordability Challenges : this is very brief. Please add more details including synthesize of tables, what are the existing articles regarding this topic, what are the main themes extracted by the relevant articles.
Author’s response 3:
The section is intentionally concise to highlight the principal cost drivers (drug price, diagnostics, infrastructure) that impede immunotherapy access. A tabulated literature synthesis would be appropriate for a systematic or narrative review; however, it would expand the article far beyond the word-limit typically allotted to Perspective manuscripts and alter its intended format.
Comment-4:
Strategies: no references at all to improve this issues. same thing, select relevant articles and create the tables, and make the theme to show the strong evidence.
Author’s response 4:
The Strategies section cites empirical and policy sources (references 27–36 in the current draft) to illustrate pooled procurement, technology transfer, and value-based pricing. In a Perspective, we prioritise conceptual clarity and policy relevance over comprehensive cataloguing of every implementation study. Creating multiple evidence tables would exceed the recommended level of detail for this format.

Round 2
Reviewer 3 Report
Comments and Suggestions for Authors
WELL ADDRESSED